# Towards Online Ageing Detection in Transformer Oil: A Review

**DOI:** 10.3390/s22207923

**Published:** 2022-10-18

**Authors:** Ugochukwu Elele, Azam Nekahi, Arshad Arshad, Issouf Fofana

**Affiliations:** 1School of Computing, Engineering and Built Environment, Glasgow Caledonian University, Glasgow G4 0BA, UK; 2Department of Applied Sciences, Université du Québec à Chicoutimi, Saguenay, QC G7H 2B1, Canada

**Keywords:** ageing, high voltage, insulator, Internet of Things, sensor, superhydrophobicity, transformer oil

## Abstract

Transformers play an essential role in power networks, ensuring that generated power gets to consumers at the safest voltage level. However, they are prone to insulation failure from ageing, which has fatal and economic consequences if left undetected or unattended. Traditional detection methods are based on scheduled maintenance practices that often involve taking samples from in situ transformers and analysing them in laboratories using several techniques. This conventional method exposes the engineer performing the test to hazards, requires specialised training, and does not guarantee reliable results because samples can be contaminated during collection and transportation. This paper reviews the transformer oil types and some traditional ageing detection methods, including breakdown voltage (BDV), spectroscopy, dissolved gas analysis, total acid number, interfacial tension, and corresponding regulating standards. In addition, a review of sensors, technologies to improve the reliability of online ageing detection, and related online transformer ageing systems is covered in this work. A non-destructive online ageing detection method for in situ transformer oil is a better alternative to the traditional offline detection method. Moreover, when combined with the Internet of Things (IoT) and artificial intelligence, a prescriptive maintenance solution emerges, offering more advantages and robustness than offline preventive maintenance approaches.

## 1. Introduction

The sole aim of power generation is to meet the electricity needs of consumers spread across homes and industries. This generated power meets different consumers at different voltage levels, and transformers are generally the equipment designed to supply the needed voltage to consumers. Transformers are expensive essential components of high voltage (HV) stations. They have an extended mean time to repair (MTTR) and enormous maintenance costs. Transformer failure can amount to the shutdown of a power station, which has serious economic consequences. Other consequences could include the loss of lives, damage to substation equipment, and environmental (ecological) effects. 

A detailed qualitative and quantitative failure mode effect and criticality analysis (FMECA) of power transformers revealed that insulation failure is a significant cause of transformer failure [1,2,3]. The transformer insulators comprise oil and paper. The transformers in most power substations require oil as a medium for cooling, arc extinction and insulation. Additionally, transformer oil acts like an information carrier, providing information on the health of the paper insulator through depolymerisation [4].

Like most HV materials, transformer oil is subject to ageing upon usage [5,6]. Insulation ageing is the gradual decrease in the dielectric strength of an insulator in operation to a complete insulation breakdown from electrical and environmental stresses. While in use, transformer oil is subject to degradation that depends on the transformer loading condition/thermal stress [7,8]. The thermal stress originates from either the windings’ copper or core iron losses. This further decomposes the oil, jointly provoking the partial discharge formation and yielding the formation of transformer oil ageing by-products (ABPs), such as moisture-dissolved gases (carbon dioxide, methane, ethane, ethylene, acetylene, propane, propylene, methanol and ethanol), acids and sludge [7,9,10,11,12]. As shown in Figure 1, Partial discharges are incipient faults that can culminate in insulation breakdown [13] with a cyclic cause and effect relationship with ageing [14]. Other partial discharge sources include voids in the pressboard, moving bubbles, and surface discharge on winding [15]. ABPs negatively impact the oil’s dielectric strength property. In addition, ageing significantly affects transformer oil’s chemical and electrical properties, such as dielectric strength (decrease), dielectric dissipation factor, DDF (increase), flashpoint (decrease) and colour (shading) [7,16,17]. The significant chemical properties altered due to ageing include acidity and turbidity, while the significant electrical properties altered include dielectric strength, the dielectric dissipation factor and resistivity [7]. 

Transformer oil insulation failure has rippling consequences on the primary purpose of power generation, the safety of personnel and the work environment, as well as enormous economic consequences. Consequently, their volume, purity, and reliability cannot be compromised. Ageing in transformer oil is primarily evidenced by interfacial tension and acidity. 

The transformer ageing detection method is classified as either intrusive or non-intrusive, destructive or non-destructive, and offline or online. Intrusive detection techniques make contact with the transformer oil as opposed to non-intrusive methods. Destructive methods alter (in the short or long run) the transformer oil properties being measured as opposed to non-destructive methods. The online detection method involves live ageing detection of the transformer oil while operating as opposed to offline ageing detection, which only involves sample collection for laboratory analysis and interpretation. 

In-service transformer oil ageing cannot be prevented [11], but it can be maintained at a lower rate through prescriptive maintenance steps using online sensors, thus avoiding catastrophic effects/outages. The report by [18] points out that aged transformer liquids can be treated by drying out, degassing, reclamation, refining and reconditioning. This will ensure that the in-service transformer functions effectively up to its expected lifespan (approx. 40 years) [11]. 

## 2. Review Methodology

This review was conducted with the research question—how can ageing detection in transformer oil systems be improved? To effectively answer this question, this paper focused on nine (9) research themes summarised in Table 1. Figure 2 and Figure 3 show the statistical plots, and Table 2 and Table 3 summarise the screening methodology and thematic references, respectively. 

This statistic did not include reference sources that did not uniquely contribute to answering the research question. Since 2019, there has been significant research interest in this subject area (see Figure 3); however, very few works have been reported on related systems (see Table 3; review of related systems), emphasising the need for additional research on the online ageing detection of transformer oil. 

## 3. Types of Transformer Oil

Mineral oil and ester oil are two common examples of transformer insulating liquids, with mineral oil having enjoyed over a century of use (compared to ester oil) and made by the fractional distillation of crude petroleum. Mineral oil is a popular insulating liquid for high-voltage transformers, jointly serving the purpose of cooling and insulation. It is low-cost and readily available [19]. However, mineral oil is toxic, non-biodegradable and potentially flammable, thus a hazard to the environment [20]. Additionally, transformer mineral oil produces more water content during use/ageing than ester oil [4]. 

Natural esters (from animal/vegetable products) and synthetic esters are the two types of ester oil currently in use [4]. Natural ester oils are more environmentally friendly and renewable, gaining increasing usage [20,21,22] at cost expense. Furthermore, ester oil resists oxidation and preserves the paper insulator better than mineral oil [23] as proved by the experiment conducted by Martins and Gomes [4]. This is because water is more soluble in natural esters than mineral oil. In addition, ester oils have better electrical characteristics (BDV) than traditional mineral oil [24]. 

When unique properties (pour point, partial discharge resistance, flammability, oxidative stability) are sought, synthetic ester oils are often used [18]. As shown in Figure 4 [25], Synthetic esters are formed from the reversible reaction of carboxylic acids and alcohol to form esters and water (esterification). The building block for synthetic esters is unlimited as there are hundreds of potential acid and alcohol building block combinations to form them [25]. Although not natural, synthetic esters have the eco-friendly properties of natural esters. 

Other alternative oils include edible coconut oil, Karanji oil [26], silicon oil, castor oil and sesame oil, which showed acceptable limits in terms of pour point, acidity, DDF and BDV, and which can be further improved by purification [27]. 

## 4. Transformer Oil Ageing Characterisation Techniques

Insulation ageing is the gradual decrease in the dielectric strength of an insulator in operation to complete the insulation breakdown from electrical and environmental stresses. Ageing affects transformer oil’s chemical and physical properties, and several ageing techniques are used to understand this. This section details some of the current ageing detection methods and their corresponding regulating standards.

### 4.1. Breakdown Voltage (BDV) Test

This electrical testing method is used to ascertain the offline age of a transformer oil sample (see Figure 5 [28]). The breakdown voltage is the voltage at which the oil sample becomes conductive (evidenced by spark) [17], usually reported in [kV/mm] units. It is also the measure of the ability of insulation to withstand electrical stress [29]. The transformer oil is placed in a test cell containing hemispherical electrodes. The BDV test is guided by the IEC 60156 and IS 6792 standards. 

According to [17], in the initial phase of electric field application, thermal agitation occurs in the oil, leading to the formation of microscopic cavities. Further application of the electric field leads to gas development and microbubble formation. Moreover, further exposure results in a gas production rate exceeding bubble formation and, consequently, the production of disruptive discharge. The voltage at which this occurs is called the breakdown voltage or the disruptive discharge voltage value [30]. An example instrument used in characterising oil samples with the BDV test is the BA75 analyser. Aged transformer oil negatively correlates with the BDV value, but a low BDV value does not necessarily imply ageing. BDV is a pointer to the severity levels of transformer oil impurity according to BS EN 60422 [31]. 

The disruptive discharge indicates an insulation failure from a breakdown voltage test instrument. The discharge completely bridges the insulation under the test, reducing the voltage between the test electrodes to zero. According to IEC 60060-1 [30], disruptive discharge may sometimes occur momentarily and is referred to as non-sustained disruptive discharge. 

The minimum acceptable BDV value is 30 kV/mm for transformers operating from 230 kV; 28 kV/mm for transformers rated between 69 kV and 230 kV; and 23 kV/mm for those rated at most 69 kV [32]. 

### 4.2. Fourier Transform Infrared Spectroscopy (FTIR)

Spectroscopy is the science concerned with investigating and measuring spectra (a plot of measured light intensity against some properties of a light example, wavelength or wavenumber [33]) produced when matter interacts with or emits electromagnetic radiation. Infrared (IR) spectroscopy studies the interaction between matter and infrared radiation based on absorbance. FTIR is useful because different chemical structures produce different spectral fingerprints. FTIR can be used to ascertain the functional groups in molecules present in oil samples [29,33] through spectral comparison and library searching. The atoms in oil vibrate with a specific frequency, which forms a peak when the IR frequency matches the atom frequency; the IR interpretation table helps identify the functional groups [17]. As oil ages, acidic and peroxide contents are formed for different chemical bonds during oxidation and thermal decomposition [31]. They have been classified as O-H (hydroxyl groups comprising alcohol and carboxylic acids) groups, C-H (methine group) and C=O (carbonyl group or carbon monoxide) functional groups. According to the FTIR analysis performed by [34], the intensity of peak absorbance of the methine and carbonyl group increases with ageing, while that of the hydroxyl group decreases with ageing. FTIR is useful for offline transformer oil characterisation.

The authors of [35] utilised a Nicolet iN10TM FTIR spectrometer to analyse automobile engine oil samples and spectral data acquired over the spectral range of 4000–500 ${\mathrm{cm}}^{-1}$ at 4 ${\mathrm{cm}}^{-1}$ spectral resolution. Each sample was scanned three (3) times to eliminate random error sources, and the average was taken for the qualitative and quantitative analysis following the ASTM E 2412-04 standard. The ASTM E 2412-04 standard covers the use of FTIR in monitoring contaminant buildup [35]. 

Figure 6 shows the result of the FTIR spectral analysis reported by [35] and ageing is evident from the plot. The cause of the ageing can also be inferred following the ASTM E 2412 standard for contamination compounds and causes. The reduction in the peak observed between wavenumbers 2955 and 2853 was attributed to the evaporation of the oil base components due to ageing, and the changes in the region of wavenumbers 1800 to 1670 were attributed to oxidation following the ASTM E 2412 standard.

FTIR has a generic application and improved signal-to-noise ratio (when compared to infrared spectroscopy) but can suffer from artefacts (features present in the spectrum of a sample outwit the sample, such as $H_{2}{O\;\mathrm{and}\;\mathrm{CO}}_{2}$) [33].

### 4.3. Dissolved Gas Analysis (DGA)

A dissolved gas analysis is a technique for characterising transformer oil to determine the beginning of the defect [21]. It is beneficial for preliminary ageing detection and localisation. Gases are produced due to the decomposition of transformer oil (mainly hydrogen gas, hydrocarbon and carbon monoxide). These gases are then collected and further analysed to determine the gases present and their percentages. Depending on the gases detected, the severity of ageing can be determined. 

DGA comprises two stages. The first stage quantifies the dissolved gases (extraction stage), while the second stage identifies the component gases and diagnoses the fault [21]. IEC BS EN 60567 is the standard for the sampling of gases from oil-filled equipment and the specification of various tools and methods for sampling and labelling [36] (stage 1), while the IEC 60599 is the standard for the interpretation of the dissolved gases using basic gas ratios [37].

N’cho and Fofana [38] listed several novel fibre optic sensors designed to detect dissolved gases (hydrogen gas $H_{2}$, carbon monoxide CO, acetylene $C_{2}H_{2}$, and methane ${\mathrm{CH}}_{4}$) in transformer oil indicative of ageing. These sensors can be integrated with data acquisition systems for online health monitoring instead of offline DGA analysis techniques. 

### 4.4. Photoluminescence (PL) Spectroscopy and Ultraviolet-Visible Spectroscopy (UV-Vis)

Photoluminescence (PL) spectroscopy is a non-intrusive, non-destructive method of probing materials. It measures the energy of light emitted during the electronic transition from an excited state to a ground state [16]. Light is directed to a sample until the sample is excited; light (photon) is released upon relaxing. PL spectroscopy measures the optical fluorescence as a function of wavelength [7]. According to [7], PL spectroscopy is more straightforward, sensitive (due to its narrow band of electronic states) and can be implemented online through available PL sensors. 

The configuration consists of a laser source, an optical lens (to converge the laser light), one cuvette for holding the oil sample, a monochromator (for selecting narrow bands of the light’s wavelength, a detector (acting as a passive transducer), an amplifier and a workstation for analysing the signal. According to [16], different experiments showed that the photoluminescence technique showed a better correlation with DDF than the UV–Vis spectroscopy technique. 

Ultraviolet–visible spectroscopy is popularly used for transformer oil condition assessment but has also faced criticism owing to its cost and poor sensitivity to fluorescence [7]. This method measures optical absorption as a function of wavelength. The configuration comprises a light source (UV and visible light), a monochromator, optical lenses, a splitter, two (2) cuvettes, a differential amplifier, and a computer workstation. One cuvette contains the pure oil sample, and the other contains the aged sample under test. The amplified result measures the UV–Vis absorbed by the aged sample. The transmittance (T) is defined as the ratio of the propagated light intensity through the sample cell ($I_{o}$) and the light intensity before propagation through cell I, expressed mathematically as:(1)$$T\;=\frac{I_{o}}{I}$$

The absorbance (A) is expressed as:(2)A =−log(T)

### 4.5. Total Acid Number (TAN)

The TAN indicates the acid concentration of the transformer oil insulator, and it strongly correlates with ageing [29,39,40]. It is otherwise referred to as the neutralisation number (NN). It is determined by the amount of potassium hydroxide (KOH) required to neutralise the acid in one gram of a transformer oil sample [5], hence expressed in mgKOH/g. According to BS EN 62021-1 [41], it is the quantity of a base, expressed in milligrams of potassium hydroxide per gram of sample, required to titrate potentiometrically a test portion in a specified solvent to obtain a pH of 11.5. Transformer oil below the marginal class of the OQIN reference is widely regarded as unsafe for continuous use and should thus be reclaimed. 

BS EN 62021-1 and BS EN 62021-2 [41,42] specify the standards for mineral oil acidity determination by automatic potentiometric titration and colourimetric titration. According to BS EN 62021-1, the test portion (transformer oil sample) is dissolved in a solvent and titrated potentiometrically using a glass-indicating electrode and a reference electrode. Potentiometric titration does not require an indicator; instead, a potential is measured. The end-point specification for BS EN 62021-1 is 11.5, and the volume corresponding to this value is reported as the neutralisation number (NN). 

An automatic pH titrimeter is recommended by BS EN 62021-1, and the acid number titrator, as shown in Figure 7, complies with this standard. 

### 4.6. Interfacial Tension (IFT)

Tension always exists at the interface of fluid phases following attraction and repulsive forces [43]. Interfacial tension is the force between two distinct phases: gas–liquid, gas–solid, liquid–liquid or liquid–solid interfaces [44]. The author of [43] defined IFT as the quantitative index of the molecular tension at a given interface, expressed as the force exerted at the interface per unit length. Streamlined to transformer oil, IFT, σ, has been defined as the force in dynes per centimetre (dynes/cm) or milliNewton per meter (mN/m) required to rupture the oil film existing at an oil–water interface [45]. The authors of [5] define IFT as the measure of the molecular attraction force between a layer of oil and a layer of water. IFT strongly (negatively) correlates with aged oil samples [46]; this implies that most characterisation techniques depend on ageing and other factors, but IFT (and NN) is a direct measure of the transformer oil sample’s age. Similarly to TAN, transformer oil samples with IFT values below the marginal OQIN reference (see Table 4) are considered unsafe for continuous use and should be reclaimed. 

For the liquid–liquid interface which characterises the transformer oil, some of the methods used to evaluate the IFT include the ring method (according to Du Noüy), plate method, rod method, drop volume method, spinning drop method and pendant drop method. Most IFT instruments mostly use the Du Noüy ring method. It involves slowly lifting a platinum ring from a liquid interface, which relates directly to the interfacial tension. Platinum surfaces are chemically inert, easy to clean, optimally wetted, and generally form a zero contact angle. The platinum ring hangs parallel to the sample and is allowed to sink into the liquid. As the ring is pulled upwards, the interfacial tension generates a force on the ring, which is measured (the maximum force on the ring when pulled out of the liquid) to evaluate the interfacial tension σ. 

Mathematically:(3)$$\sigma =\frac{F_{max}}{2\pi D}\;\cdot \;C$$σ represents the interfacial tension, *F* is the force acting on the platinum ring, *D* is the diameter of the ring and *C* is the correction factor (taking into account the additional volume of liquid extracted together with the ring). Mathematically, *C* can be evaluated as:(4)$$C=0.725+\sqrt{4.014\times 10^{-4}\times \frac{\gamma }{\Delta \rho }+0.01287\;}$$
γ represents the interfacial tension without correction in mN/m, and Δρ represents the density difference between water and the insulating liquid at measuring temperature in ${\mathrm{gcm}}^{-3}$. The Du Noüy ring method is recommended by the ASTM D971 and BS EN 62961 standards. Automatic tensiometers calculate the σ and *C* values. According to BS IEC 62961 [47], automatic tensiometers can overcome the film destructive nature of the Du Noüy ring method by automatically determining the maximum force value on the ring for detachment from the film before detachment and reverse the platform movement promptly prior to detachment without tearing the film. This allows for accurate, time-consistent serial measurements. The ring method has some limitations, including the need for a vibration-free location, a delicate ring, a non-flammable laboratory and training to be undertaken. 

The Wilhelmy plate method is also commonly used for offline IFT measurement using a platinum plate. Mathematically, the interfacial tension using this method is evaluated using:(5)$$\sigma =\frac{F}{L\cos\theta }$$
where σ represents the interfacial tension, *F* represents the force acting on the plate, *L* is the wetted length and θ represents the contact angle. The contact angle measures the degree of wettability of the plate. For platinum plates, this value is near zero, emphasising its hydrophilic property. The advantage of the Wilhelmy plate method over the ring method is that the plate method for the IFT measurement is static, making it suitable for recording changes, unlike the destructive ring method [48]. To measure the force, *F*, the plate is attached to the force sensor of a tensiometer. The rod method for the IFT measurement follows the same principle: a cylindrical rod is used instead of a plate. The rod method is more applicable for lower resolution measurements because the wetted length is smaller when compared with the wetted length for plates. 

Other non-forced methods for IFT measurement include the pendant drop method (an optical method), the drop volume method and the spinning drop method [49]. Force tensiometer instruments measure the forces exerted on a probe positioned on a liquid–liquid interface. Usually, the probe is connected to a very sensitive balance that reports the interfacial tension value. Force measured depends on the size and shape of the probe, the contact angle between the probe and the liquid, and the interfacial tension between the liquids. To overcome measurement bias, sufficient care must be taken to ensure that the force measured is solely determined by interfacial tension. The Sigma 702ET instrument is a specialised instrument for measuring offline laboratory standard interfacial tension in accordance with ASTM D971 and IEC 62961 standards.

BS IEC 62961 [47] specifies that to obtain a value that provides a realistic expression of the real interfacial tension (for the ring method), a measurement must be made after a surface age of approximately 180 s. Surface age is defined as the period from the beginning of the production of a surface or interface to the time of the observation or measurement. BS IEC 62961 also specifies the ring cleaning methods, vessel preparation technique, and IFT determination and reporting procedure.

## 5. Classification of Service-Aged Insulating Oil

The Myles Index Number or Oil Quality Index (OQIN) provides a metric for transformer oil classification into seven categories. The OQIN value is the quotient of the IFT value and the NN value. The OQIN index can be used for offline and online oil classification. Another index suggested by the authors of [29,46] is the ratio of the dielectric dissipation factor to the FTIR transmittance measured at 1710 ${\mathrm{cm}}^{-1}$ (DDP/T) index. The DDP/T index is zero-based but currently limited to offline analysis. 

## 6. Accelerated Thermal Ageing

Studying the complete degradation cycle of transformer oil (early life, mid-life, end-of-life) requires fifteen (15) to fifty (50) years of observation of natural ageing, data collection and analysis [50]. This is more than the duration of most research projects (three to six years). Therefore, studying the ageing process and its impact on the dielectric requires an accelerated simulation of the natural ageing process, otherwise known as accelerated ageing techniques. Accelerated ageing involves combined electrical and environmental stresses similar to those in the field, following specified standards.

Accelerated thermal ageing helps to reveal critical degradation properties (physical, chemical and dielectric) in transformer oil due to the ageing by-products (ABPs) generated [7]. The half-life rule is consistently used to conduct accelerated thermal ageing [7] and states that transformer oil’s ageing rate doubles as the operating temperature increases by 7 °C. The base operating temperature for transformer oil is between 55 °C and 65 °C [52]. The ageing accelerating factor (AAF) and time factor are calculated using:(6)$$\mathrm{AAF}\;=\frac{\mathrm{ATT}\;-\;\mathrm{BOT}}{7}$$
(7)$$\mathrm{TF}\;=2^{\mathrm{AAF}}$$

ATT stands for accelerated thermal temperature; BOT stands for base operating temperature; and TF stands for time factor. This implies that for an accelerated thermal temperature of 120 °C, assuming a base operating temperature of 60 °C, one (1) day of ageing in the heating oven will be equivalent to 380 days of practical ageing in the field (see Figure 8 [7]). Standards mostly govern accelerated thermal temperatures. An exception would be when modelling a unique operational condition within the transformer, such as a hot spot [51]. Different samples can be prepared based on the half-life rule representing different thermally aged field samples. ATT values usually range from 100 °C to 160 °C. Aged samples not expressed in field operation days (using the half-life rule) are expressed in hours of heating. Typical examples include hours in steps of 500 (500, 1000, 1500, 2000, 2500, 3000, etc.). 

A better setup that replicates a power transformer more closely was reported by Bouaicha, Fofana and Farzaneh [53]. Their setup included metallic catalysts (zinc, copper, aluminium and iron) to simulate the metallic components in the transformer (windings and core). Oil-impregnated pressboards were used to simulate the kraft paper insulation, and silica gel was rightly positioned to simulate the transformer breathing mechanism. The half-life rule could have been used to replicate the thermal degradation even more closely. The ASTM D1934-95 standard provides the specification for oxidative ageing using two procedures (without a metal catalyst and with a metal catalyst) [54].

## 7. Cross-Capacitive and Fibre Optic Ageing Detection Sensors 

Machine learning IoT solutions will improve HV insulation condition monitoring [55] and sensors play an essential role in generating data for machine learning diagnostics. 

### 7.1. Cross-Capacitive Sensor

The authors of [56] reported the effectiveness of parallel plate capacitive sensors with thin hydrophilic porous oxide metal film as moisture content detection sensors. However, one of the setbacks highlighted with the parallel plate capacitive sensor is its instability due to variations in the geometric parameters, such as a gap shift, bending of electrodes, or even overlap of electrodes. Therefore, the cylindrical cross-capacitor proposed by Thompson and Lampard is used to overcome these geometric issues. It comprises four (4) identical cylindrical electrodes separated by gaps. The capacitance between the diagonal opposite pair of electrodes is called the cross-capacitance.
(8)$$\;C_{c}=\left(\frac{\in \times \log_{e}2}{\pi }\right)\times l$$
$C_{c}$ represents the cross-capacitance; ∈ represents the dielectric permittivity; and *l* represents the length of the electrodes.

In the simulation experiment (using ANSYS Maxwell Version 15.0) performed by [57], the cross-capacitance value did not vary with changes in the structural design of the cross-capacitor (circular or square electrode). Insignificant changes with different uniform insulation gaps were also reported. The capacitance value depends on the length of the electrode and the dielectric medium within the electrode. This is an improvement over parallel plate capacitors that depend on the distance between the plates, the area of plates and the permittivity of the dielectric. Furthermore, since the electrode lengths are fixed, the capacitance depends solely on the oil dielectric. As a result, the cross-capacitive sensor is more robust and suitable for transformer applications. It is also temperature-independent [57].

Cross-capacitive sensors have been reported to be effective in measuring moisture concentration, dielectric constant, resistivity, humidity and metal detection. However, changes are usually made to the electrodes’ sensing film or geometric arrangement. 

Rahman, Islam, Khera and Khan [56] designed a novel cross-capacitive sensor applicable for contact measurements of moisture and 2-FAL. An AD 7150 capacitance-to-voltage sensor was utilised to infer the capacitance of the dielectric medium. The results showed a linear response to moisture and different 2-FAL concentration levels. 

### 7.2. Fibre Optic Sensor

Fibre optic sensors have the following advantages over most other sensor types: electromagnetic interference immunity, small size, lightweight, high sensitivity, large bandwidth, extreme environment adaptability, explosion-proof, resistance to ionisation, resilience, and suitable for remote and distributed sensing (see Table 5) [38,58,59]. They can be applied in monitoring physical parameters, such as strain, temperature, pressure, humidity, refractive index and HV insulation ageing monitoring (by inference). Optical fibres are characterised by a uniform refractive index used in different configurations to measure the optical absorption, fluorescence refractive index, pressure and strain [38,60]. As shown in Figure 9, optical fibres are generally classified by the materials used (plastic, glass, polymer, silicon fibres), refractive index (step index or graded index), and mode of light propagation (single-mode or multimode). Multimode transmission has fibre diameters (50 and 60 μm) much larger than the wavelength of the light source, giving the light many paths to take as it propagates through. On the other hand, single-mode transmission has fibre diameters (9 μm) almost equal to the transmitted light wavelength, allowing only one wavelength to propagate through. 

Different sensitivity responses are expected based on the choice of material, component refractive index and mode of propagation. For example, an experiment conducted by [61] for ageing monitoring of an industrial liquid coolant showed that silicon fibres exhibit more efficient behaviour than polymer fibre (when used for acidity monitoring) in terms of stability and repeatability. However, polymer fibres have better sensitivity, biocompatibility and bending tolerance [59]. 

An optical fibre consists of the core, the cladding and the coating. The core is the light transmission area of the fibre that can be of either glass or plastic material. The cladding provides a lower refractive index at the core interface to allow for a reflection within the core for light transmission through the fibre based on the principle of total internal reflection [59]. Therefore, the cladding ensures a decrease in scattering loss and offers additional protection to the core. The coating offers environmental and mechanical protection to the optical fibre core and is mainly made from plastic [59]. 

The general structure of an optical fibre sensor is composed of an optical source (or transmitter), the sensing element and the optical detector. The sensing area is developed by uncladding a choice portion of the fibre optic cable. The optical source can be either a light-emitting diode (LED) or a laser diode. LEDs are generally suited for multimode fibres and are characterised by low power, less bandwidth and a maximum throughput of 1 Gbps. Laser diode sources offer more advantages than LED sources at the expense of cost. Laser diodes are faster, allow single-mode or multimode transmission (Fabry-Perot lasers) and can achieve a throughput of at least 10 Gbps. Distributed feedback (DFB) lasers and vertical cavity surface-emitting lasers (VCSELs) do not enjoy the versatility of Fabry–Perot lasers as they are only suited for single-mode fibres and multimode fibres, respectively. 

Optical fibre sensors are classified as intrinsic and extrinsic sensors [59]. Intrinsic fibre optic sensors have a sensing area within the fibre optic cable (the fibre is the sensor), which is usually achieved by uncladding a portion of the fibre optic cable. The developed sensing area makes direct contact with the material to be measured, and a property of light is modulated, which infers the input. The fibre optic cable is a light waveguide for extrinsic fibre optic sensors to the external sensor. The intrinsic sensing mode is more common with most optical fibre sensors. For the intrinsic sensor configuration, the measurand alters light wave parameters, such as wavelength, phase, intensity and polarisation, which can be used to infer the measurand quantity. For this reason, fibre optic-based sensors (OFS) are either phase-modulated, intensity-modulated, wavelength-modulated or polarisation-modulated [59]. 

Intensity-modulated OFSs are common for intrinsic fibre optic configurations. The sensing principle is based on the attenuation of the optical signal induced by the monitored parameter [59]. Following the loss of signal power, which characterises intensity-modulated OFSs, higher optical signals are recommended (multimode fibres). Intensity-modulated OFSs, although simple to implement and adaptable for distributed instrumentation, require standardisation and source referencing to impede false interpretations. Optical intensity attenuation is mainly achieved through microbending and evanescent field absorption. Microbends are bends too small to be detected by the human eye when external pressure is applied to a defined portion of the fibre optic sensing area. Optical intensity attenuation occurs during microbending when propagated guided signals radiate into non-guided modes. According to [59], the microbend affects the critical angle that allows the propagated signal to remain in the guided mode. Microbend OFSs have found applications in strain, pressure, temperature, vibration, humidity and pH measurements [59]. Evanescent field sensors rely on intensity loss due to the uncladding of a portion of the fibre optic cable. This ensures that the total internal reflection does not lead to signal loss which depends on the refractive index of the sensing medium. For transformer oil applications, the refractive index of aged samples varies proportionally to the age of the oil samples. Polishing the sensing area (superhydrophobicity) of evanescent field sensors can free the sensing area from impurity clogging when in contact with test samples. 

A single-mode fibre-sensing configuration comprises an optical fibre, a sensing component, a photodetector and a light source, with the fibre acting as the transmission medium and the sensing element. However, the single-mode–multimode–single-mode (SMS) configuration is reportedly more effective, and the multimode section of the configuration is used as the sensing element. The single-mode step-index polymer optical fibre is reportedly adequate for monitoring transformer oil breakdown [38]. The authors of [58] used a step-index multimode polymer optical fibre to design a water-quantity sensor for transformer oil application. The sensing area was determined by mechanically uncladding a portion (2 cm) of the multimode fibre. The authors of [58] summarised the design process for a multimode polymer step index sensor generally applicable for other related measurements. The setup is summarised below. 

#### Evanescent Wave Absorption Principle for Online Ageing Detection 

The speed of light changes as it moves between media, which causes refraction. Light travels faster in a vacuum than it does in any material. The refractive index of a material (water, transformer oil) is a measure of the change in the speed of light as it passes from a vacuum (or approximately air) into the material; it is mathematically expressed as:(9)$$n=\frac{V_{1}}{V_{2}}$$
where *n* represents the refractive index of the material; $V_{1}$ represents the speed of light in a vacuum; and $V_{2}$ represents the speed of light in a material. The refractive index is inversely correlated with the speed of light through the medium. A higher refractive index implies a lower speed of light. 

The evanescent field absorption-based configuration is a widespread intrinsic sensing technique [32]. A portion of the fibre optic sensor is uncladded, forming the sensing area. The parameter to be detected affects the refractive index of the uncladded portion and prevents optical waves from leaking into the fibre optic waveguide [32] and consequently diminishes some properties of the wave reaching the fibre output. 

Mathematically [32]:(10)$$\mu =\frac{P_{clad}}{P_{clad}+P_{core}}\;≅\;\frac{4}{3V}$$
μ is the modal fractional power, $P_{clad}\;\mathrm{and}\;P_{core}$ represent the powers carried on the core and cladding, respectively, and *V* is the fibre’s normalised frequency. The normalised frequency is a unitless quantity expressed as: (11)$$V=\frac{2\pi a}{\lambda }\;\sqrt{n^{2}{}_{core}-n^{2}{}_{clad}\;}$$
*a* is the fibre core radius, λ is the wavelength of the optical source, and $n_{core}\;\mathrm{and}\;n_{clad}\;$represent the refractive indices of the core and cladding, respectively. With constant source wavelength, core diameter, and core refractive index, $P_{clad}$ becomes proportionally related to the clad refractive index, $n_{clad}$ (combining Equations (9) and (10)). The uncladded region (sensing area) makes direct contact with the ageing transformer oil sample, and so $n_{clad}$ becomes replaced with $n_{oil}$, and $P_{clad}$ is replaced by $P_{oil}$, representing the power lost to the transformer oil owing to changes in the refractive index of the oil.

Ageing increases the refractive index of the transformer oil and consequently increases the power lost to the transformer oil. This power will not get to the fibre optic output, thus defining a sensing area (uncladded) that responds to the transformer oil ageing behaviour.

## 8. Superhydrophobicity and Online Ageing Detection

Transformer oil ageing may impact intrusive ageing detection technology due to ABPs clogging the sensing area, thus affecting the detection and consequently defeating the overall aim of the online ageing detection system. Superhydrophobicity (or ultrahydrophobicity) technology has been developed for outdoor insulators exposed to accretion and icing [62] as well as solar panels [63]. A superhydrophobic surface (first seen in lotus leaves) is difficult to wet, possesses self-cleaning characteristics, and has a contact angle greater than 150 degrees and a sliding angle (or contact angle hysteresis) less than ten (10) degrees (see Figure 10) [64,65,66]. Superhydrophobicity is a nature-inspired technology from lotus leaves, rice leaves, mosquito eyes, butterfly wings, rose petals, snail shells and fish scales [67]. 

The contact angle is measured statistically and dynamically when the liquid resides on the surface and relates to the surface tension between the different interfaces. Young’s model relates the contact angle with the surface tension of the different interfaces for smooth (or refined) surfaces:(12)$$\cos\theta =\frac{\gamma_{sv\;}-\gamma_{sl}}{\gamma_{lv}}$$
where θ represents the contact angle (or Young’s contact angle); $\gamma_{sv\;}$represents the surface tension between the solid–vapour interface; $\gamma_{sl}$ represents the surface tension between the solid–liquid interface; and $\gamma_{lv}$ represents the surface tension between the liquid–vapour interface. The Wenzel model and the Cassie–Baxter model relate to rough surfaces. The Wenzel model is expressed as:(13)$$\cos\theta =\frac{\gamma_{sv\;}-\gamma_{sl}}{\gamma_{lv}}\cdot r\;$$
where *r* represents the roughness ratio (*r* = 1 for a smooth surface; and *r* > 1 for a rough surface). 

Creating a superhydrophobic surface involves several approaches, such as chemical vapour deposition, wet chemical reactions, electrochemical deposition, the layer-by-layer method, self-assembly, electrospinning, plasma treatments, sol-gel, etc. Studies [68,69,70,71,72,73,74,75,76,77] have aimed at providing low surface energy (using fluorine or silicon-containing molecules; for rough surfaces) and high micro-nanostructure surface roughness (by the introduction of pores or microstructure; for non-rough surfaces) [63,78]. Surface energy is associated with the intermolecular forces between two media interfaces. 

Most works on superhydrophobic coating are applied to outdoor insulators, biomedical instruments [65] and non-intrusive measurement technology. However, more research is needed to assess the potency of superhydrophobicity in guaranteeing the reliability of intrusive online ageing detection technology for transformer oil applications prone to ABPs. This will involve applying a superhydrophobic coating to the sensing area of the uncladded optical fibre configuration, ensuring that the optical properties are not significantly impacted. The reliability can be tested by subjecting the superhydrophobic optical fibre sensor to different aged samples and measuring the precision with historical data. While the presence of ABPs is expected to impact the precision of a test conducted without superhydrophobic coating, the precision result should be much better with superhydrophobic coating.

## 9. Machine Learning Models for Online Ageing Detection

Machine learning models (regression and classification) are instrumental in achieving online ageing detection for transformer oil. Regression models are required for predicting key ageing properties, such as interfacial tension (IFT), acidity (TAN), turbidity, decayed dissolved particles (DDP), dielectric dissipation factor (DDF), etc., from the sensor’s output variable. Classification models are required to distinguish the different levels of severity using the oil quality index (OQIN) value. This enables predictive maintenance options, alerting operators of dangerous thresholds (class D to G) and providing initial online recommendations. Some examples of machine learning regression models relevant for online ageing detection include linear regression, nonlinear regression, support vector machine (SVM) regression, decision trees regression, and shallow/deep neural network regression models. Some examples of machine learning classification models relevant for online ageing detection include logistic regression, decision trees, k nearest neighbour (KNN), support vector machines, and shallow/deep neural networks. The predictor variables will be the sensor’s output variable(s), while the target variables will be any of the desired ageing properties’ variables (for regression models) or the severity level of ageing in classes or colour (for classification models). These models can be developed and evaluated using python libraries, MATLAB and R. 

The regression model is one of the most familiar models that maps a target variable to one (simple linear regression) or more (multiple linear regression) predictor variable (s) using a linear function. It works with the assumptions of linearity, independence, homoscedasticity and normality between the predictor and target variables. 

The simple regression equation is specified as:(14)$$\hat{y}={\hat{b}}_{0}+{\hat{b}}_{1}x+\in$$
where $\hat{y}$ is the estimated dependent variable, ${\hat{b}}_{0}$ is the constant term, ${\hat{b}}_{1}$ indicates the sensitivity of *y* to *x*, and ∈ is the regression error [79].

The corresponding multiple regression model is specified as:(15)$$\hat{y}={\hat{b}}_{0}+{\hat{b}}_{1}x_{1}+{\hat{b}}_{2}x_{2}+\ldots +{\hat{b}}_{k}x_{k}+\in$$
where $\hat{y}$ is the estimated dependent variable, ${\hat{b}}_{0}$ is the model constant, ${\hat{b}}_{1}$, ${\hat{b}}_{2}$, … ${\hat{b}}_{k}$ are the estimated parameter values and model sensitivities, and ∈ is the regression noise [79].

A more detailed overview of the linear regression model covering model specification, model estimation, maximum likelihood, statistical inference, regularisation and linear model diagnostics can be found in [80]. 

Regression and non-binary classification models can be constructed using decision trees and neural networks. Decision trees are tree-like graphs in which each leaf node denotes the result of cumulative decisions, and each branch node denotes an option among numerous options [81]. Artificial neural networks (ANN) are a supervised machine learning technique mimicking biological networks using connected layers. A detailed review of decision trees and artificial neural networks can be found in [82,83].

IoT data collection can produce inconsistent, sparse and noisy data, reducing the data analysis’ confidence and uncertainty. The dependability of data analysis is increased by managing and quantifying model uncertainty. Additionally, it strengthens confidence in the model’s judgments [84]. A detailed review of uncertainty quantification for machine learning models and deep learning models highlights the use of Bayesian neural networks and Bayesian physics informed networks for deep learning uncertainty quantification, as well as gaussian process regression (GPR) and physics-informed neural networks for traditional machine learning [85].

## 10. IoT and Online Ageing Detection

The Internet of Things (IoT) refers to the technology of interconnection of human assets (also known as things) via the internet using sensors to make informed decisions, minimising human intervention [86]. For example, online ageing detection can be advanced when integrated with IoT technology. The IoT helps prevent considerable person-hour loss to collect and centralise online instrument readings. The IoT provides a more straightforward solution to this challenge, including an extra layer for advanced data analysis and reporting. The things within an IoT network must stay connected regardless of their movements. Some IoT enabling technologies include low-cost/low-power sensor technology, cloud computing platforms, artificial intelligence technologies, etc. 

According to Li et al. [86], an IoT service-oriented architecture (SOA) comprises a sensing layer, network layer, service layer and interface layer. The sensing layer primarily consists of the sensors (RFID tags, barcodes, etc.) [87], data acquisition devices and protocols (NI DAQ); the network layer consists of wired or wireless connections (Bluetooth, WIFI, loraWAN, GSM&Sigfox, Satellite); the service layer meets the user’s demands; and the interface layer allows for the interaction between the user and the application. Therefore, it is recommended that IoT systems be designed to provide for the extensibility, scalability, modularity, interoperability and reliability of the implemented AI models. An IoT platform (ThingSpeak, ThingWorx, Google Cloud, Oracle IoT, Microsoft Azure IoT, etc.) integrates these sensor data from different devices through appropriate security channels and includes an extra analytics layer for real-time interpretation and implementation.

IoT systems suffer from some challenges, mainly sensor and data security challenges. There are three (3) challenges with sensors: sensor accuracy to detect the necessary data, environmental/operational impact on sensor accuracy [88], and sensor ageing. All these affect the fidelity of IoT data.

There is a need to select and locate the best sensor that can detect the unique functional characteristics of an asset. One sensor may not satisfy this condition and thus the need for a combination of sensors at the expense of cost. If possible, sensors should be shielded from operational and environmental conditions affecting their accuracy. Sensor ageing or sensor deterioration is the sensor’s sensitivity reduction owing to prolonged usage or/and the impact of external influences on the sensor. The authors suggest the periodic recalibration or retrofitting of sensors to preserve the accuracy of the IoT data. In addition, the authors suggest a predictive algorithm to detect the fidelity of these sensors while idle or in operation. This algorithm should safely notify the user or operator to recalibrate or retrofit the sensor. 

Security challenges abound and pose a significant threat, especially when connected to cloud technologies [89]. A feasible solution will include measures to detect cyber attacks in hacking or phishing and alert experts when necessary. Additional layers of security can be added to make different cyberattack activities impossible. Edge computing helps to solve security challenges by allowing computing and data storage to occur at the edge of the internet instead of the cloud. 

The IoT can be integrated to track ageing and make proactive decisions to ensure optimal transformer performance for the online ageing detection of transformer oil. Although there has not been significant research output in this area for transformer oil, the authors suggest adopting Figure 11 as the framework for online ageing detection in transformer oil. An intensity-modulated optical fibre sensor, cross-capacitive sensors, and ageing gas sensors (examples include hydrogen gas sensors and carbon dioxide gas sensors) can be utilised for the sensing layer. National Instrument NI USB 6008/6009, Arduino microcontroller, and Raspberry Pi are some examples of data acquisition devices for ageing data. ThingSpeak, ThingWorx, Google Cloud, Oracle IoT and Microsoft Azure IoT are examples of IoT platforms that can provide the network, services and interface layer. 

Thingspeak has reportedly succeeded in several IoT applications and can be applied to transformer oil ageing systems. Thingspeak is an open-source IoT platform [90] compatible with MATLAB (for AI capabilities), microcontrollers and DAQs. It has been utilised for real-time room condition monitoring systems, water monitoring systems and analytics, air quality monitoring systems, traffic control systems, and transformer subsystems [90,91,92,93,94,95,96]. When used with MATLAB (for example), the channel ID and API key are inputted to enable sensor data to be read with the help of the ESP8266 WIFI module and Arduino microcontroller. In addition, the MATLAB data analysis toolkit, Simulink and Simscape can be employed for data cleaning, analysis, predictive model development, and the design of frontends and user interfaces. 

IoT has applications in healthcare, aviation, intelligent homes, agriculture, food processing, security surveillance, reliability engineering, pharmaceutical, retail and logistics, recycling and entertainment [87,89]. This can be integrated for the online ageing detection of transformer oil.

## 11. Example of Related Systems

### 11.1. A Non-Destructive, Non-Intrusive Design Using an Antenna 

Rohit, Sisir and Marley [9] implemented a non-intrusive, non-destructive technique for the condition assessment of transformer oil as shown in Figure 12. A horn antenna was placed over an open vessel in an anechoic chamber (a room designed to stop sound reflections or electromagnetic waves) to prevent interference, filled with oil to study its characteristics. The anechoic chamber was maintained at a temperature of 20 °C and 50% humidity using a temperature controller and a dehumidifier; this is because the frequency response of the insulating liquid is sensitive to temperature and humidity variation. The parameters monitored to detect ageing were moisture content (MC) and total acid number (TAN). Laboratory samples were initially used before extensions were made to actual in-service transformer oil samples for validation. Accelerated thermal ageing and oxidation ageing approaches were used to study transformer insulation’s early, midlife, and afterlife properties with several ageing samples. The frequency response plot clearly showed the degree of ageing present in the transformer oil. A linear fit was derived from the plot of the reflection coefficient, $\left|S_{11}\right|$ against the moisture content and the TAN. Guranteeing constant humidity and temperature for an in situ transformer will involve additional costs to implement new controlled designs. This solution is also limited by the need to significantly re-mechanise existing transformer designs to accommodate the anechoic chamber. An evanescent wave absorption intrusive fibre optic solution offers a better solution in terms of cost, online adaptability, simplistic design and immunity to external interference. 

### 11.2. An Intrusive Ageing Detection Design Using a Cross-Capacitance Sensor

Rahman et al. [57] used a cross-capacitive sensor to predict the dielectric constant of a transformer insulating oil with a known contamination level of 2-Fal, and the combination of moisture and 2-FAL, mineral oil, synthetic ester, soybean oil and rapeseed oil. The oil cell measured the dielectric constant of the various solutions. The plot of the dielectric constant against different concentrations for the oil samples shows linear characteristics. The results were reported for the simulated cross-capacitance of various transformer oil concentrations. The sensor was calibrated, and a linear calibration result was reported with a maximum error of 0.86%, 0.33%, 0.91% and 0.64% for mineral oil, synthetic ester oil, soybean oil and rapeseed oil, respectively. The sensor was validated for online ageing detection using cross-capacitance technology. Cross-capacitance intrusive ageing detection technology offers good repeatability but is prone to electromagnetic interference. The moisture content affects the permittivity, making it difficult to discriminate between ageing and moisture content detection when using this solution. 

### 11.3. An Intrusive Ageing Detection Design Using Fibre Optic Technology

The authors of [10] as shown in Figure 13 used a single-mode–multimode–single-mode (SMS) fibre optic configuration to track the ageing of transformer oil. Four samples of palm-based transformer oil were used (S1, S2, S3, S4), with S1 representing the pure sample and S2 to S4 representing samples of varied age. The lightwave refractive index was used for the calibration; changes in transformer oil’s age affect the lightwave refractive index (RI). Two ageing characterisation techniques were used to determine the age of the unknown samples: the breakdown voltage test (BDV, with 25 kV) and the UV–Vis spectrum using appropriate standards. 

A light source (WSL-100) was injected through the SMS fibre optic configuration, and the MS9740A optical spectral analyser was used to display the plots (wavelength, absorption and refractive index). The resonant wavelength shift plot against the refractive index showed linear behaviour. 

This configuration can be improved for real-time ageing detection with a photodetector, DAQ device and the IoT.

## 12. Discussion, Conclusions, and Future Work

### 12.1. Discussion and Conclusions 

The need for transformer insulating oil, which provides insulation, cooling and arc extinguishing functions, has led to increased research in transformer oil insulation for improved insulation performance/reliability, cost, and eco-friendliness. This has led to the transition from mineral insulating oil, which is non-biodegradable and potentially flammable, to other choices, such as karanji oil, silicon oil, castor oil, sesame oil and particularly natural and synthetic ester oil, which trade cost for eco-friendliness, improved electrical characteristics, and an expanded remaining useful life, RUL, for paper insulation. 

Literature studies revealed that transformer oil is subject to ageing upon usage even when the transformer is operated within the recommended limits. Furthermore, transformer oil ageing affects the insulation characteristics, which has severe consequences (economic and fatal) when left undetected or unattended. Traditionally, offline detection methods have been used to characterise the health state of transformer oil in use. These methods were developed based on research outputs that showed that transformer oil ageing affects the oil’s physical, chemical and electrical properties. Some of the identified offline detection methods currently in use include the breakdown voltage test (BDV), spectroscopy methods (FTIR, PL, UV-Vis, turbidity, colour), a dissolved gas analysis, and acidity quantification against a reference (OQIN), which classifies transformer oil’s condition from good to disastrous. Literature studies conducted reviewed that interfacial tension (IFT) and total acid number (TAN) are the two (2) characterisation techniques that reflect transformer oil ageing more accurately, unlike other characterisation techniques that are functions of other parameters that may not correlate with ageing. Consequently, the OQIN index is a TAN and IFT values function. 

### 12.2. Future Work 

The offline ageing detection technique involves sampling, which will require extra training and poses a risk of contamination during the transportation of samples, thus leading to research on online ageing detection, which promises to offer improved, reliable, and safe detection, especially when integrated with the IoT. Sensors with high correlative values with ABPs are being developed and improved for online ageing detection, such as hydrogen gas, cross-capacitive and fibre optic sensors. Fibre optic sensors have the advantage of electromagnetic interference immunity, are lightweight, explosion-proof, and have improved sensitivity. However, there is no significant research output on determining the OQIN of transformer oil online using fibre optic sensors. In addition, superhydrophobicity can be applied to intrusive fibre optic sensors based on evanescent field absorption to improve the detection reliability of these sensors subject to measurement error from APBs clogging the fibre optic sensing area. Integrating fibre optic sensors for transformer oil ageing detection with the IoT and artificial intelligence (for online OQIN determination and classification) will result in a prescriptive maintenance solution which offers more advantages and robustness than traditional preventive (offline) maintenance approaches. Specifying the machine learning model’s uncertainty increases confidence in the model’s judgement, and this can be accomplished using a gaussian process regression (GPR) and physics-informed neural networks for traditional machine learning.

## Figures and Tables

**Figure 1 sensors-22-07923-f001:**
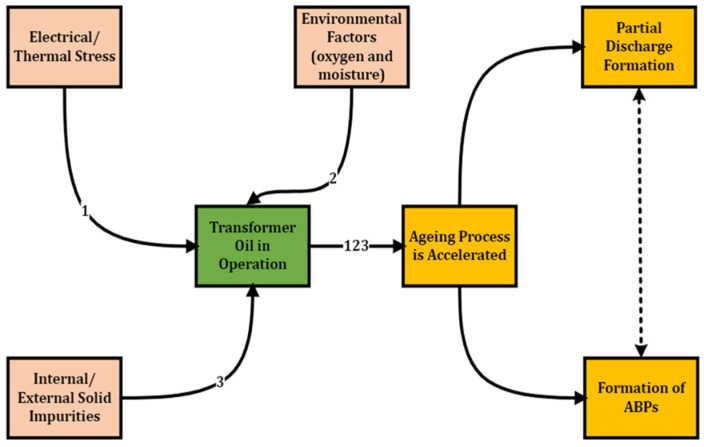
Degradation Mechanism of Transformer Oil.

**Figure 2 sensors-22-07923-f002:**
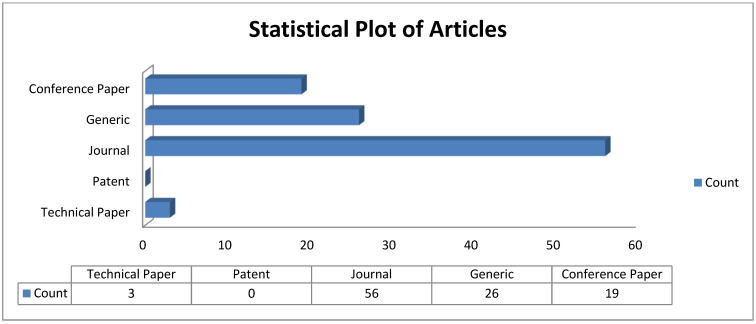
Statistical Plot of Articles.

**Figure 3 sensors-22-07923-f003:**
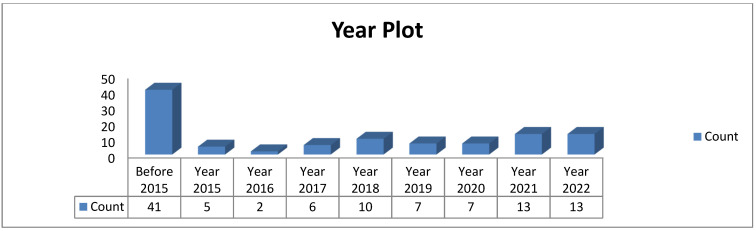
Year Distribution of Articles.

**Figure 4 sensors-22-07923-f004:**
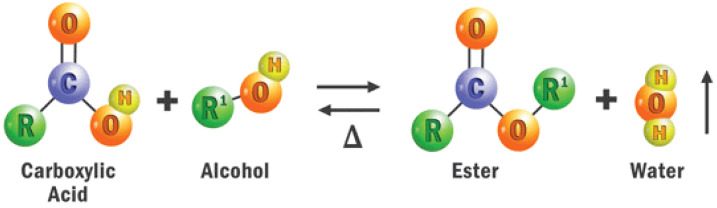
Reversible esterification reaction [25].

**Figure 5 sensors-22-07923-f005:**
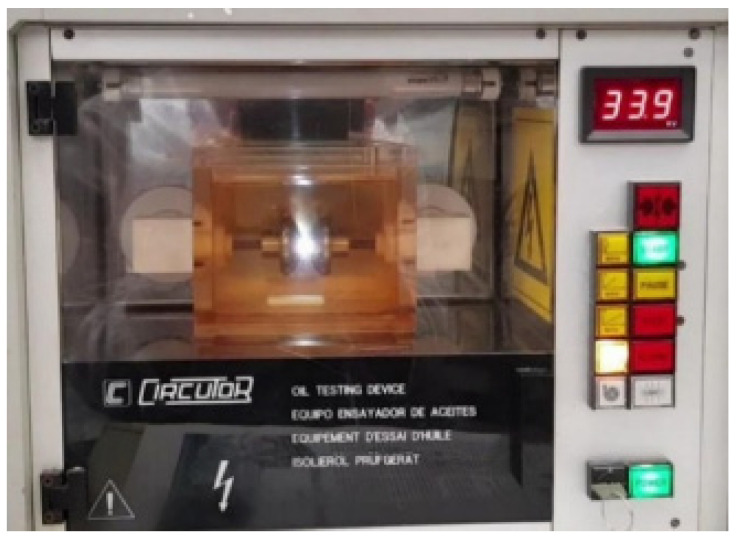
Breakdown voltage tester [28].

**Figure 6 sensors-22-07923-f006:**
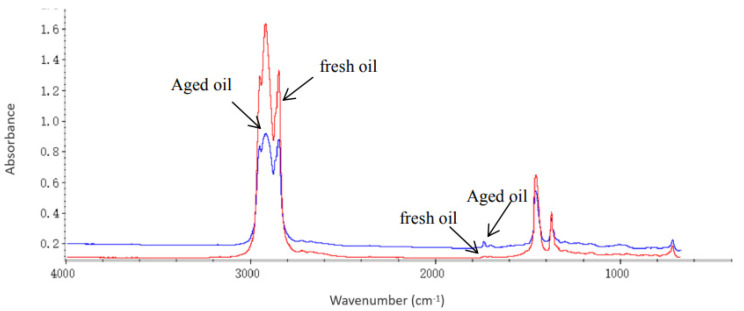
FTIR Spectral Analysis of Fresh and Aged Oil [35].

**Figure 7 sensors-22-07923-f007:**
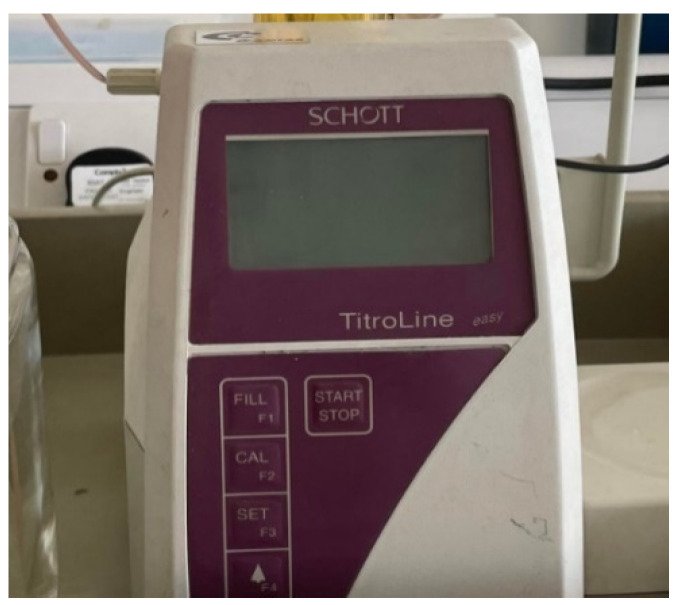
Acid number titrator in GCU Lab.

**Figure 8 sensors-22-07923-f008:**
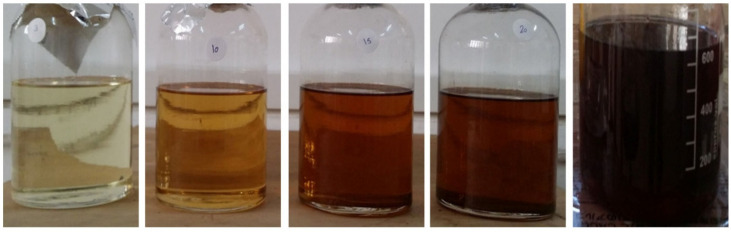
Different aged samples (L-R 3-Day, 10-Day, 15-Day, 20-Day, and 30-Day) [7].

**Figure 9 sensors-22-07923-f009:**
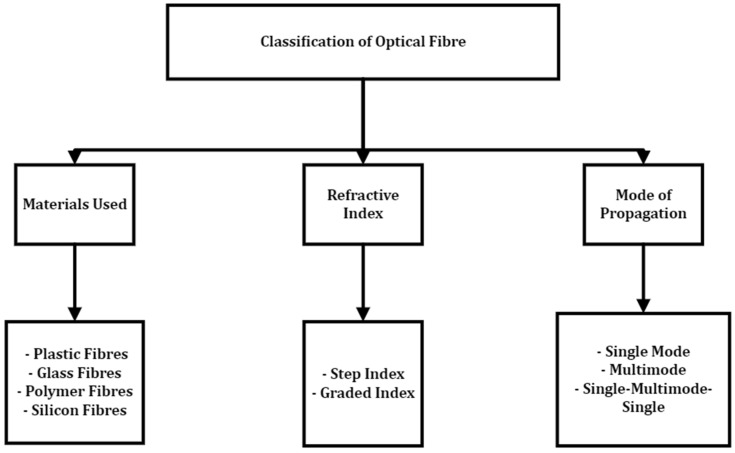
Optical fibre classification.

**Figure 10 sensors-22-07923-f010:**
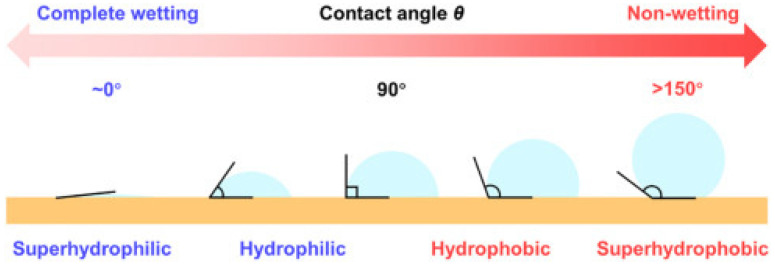
Classification of wettability of a surface [66].

**Figure 11 sensors-22-07923-f011:**
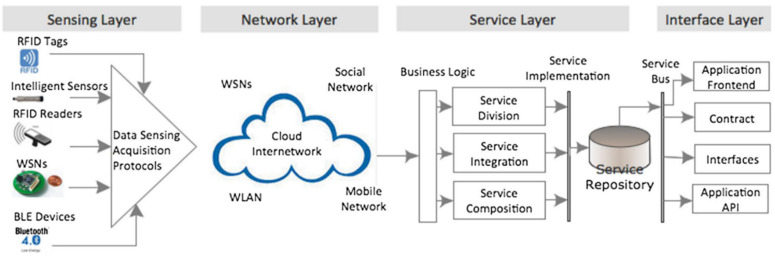
IoT service-oriented architecture [86].

**Figure 12 sensors-22-07923-f012:**
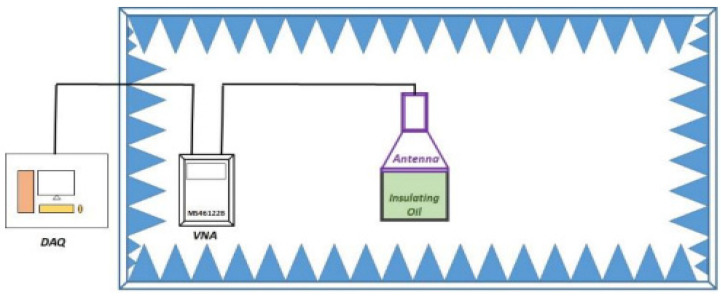
Antenna design setup [9].

**Figure 13 sensors-22-07923-f013:**
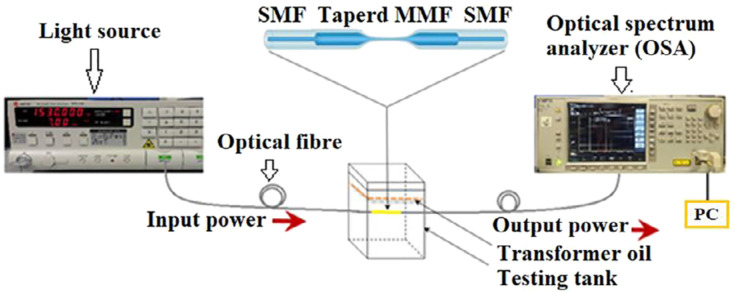
Experimental configuration [10].

**Table 1 sensors-22-07923-t001:** Review themes.

Themes
1. Review of various transformer oil types
2. Review of transformer ageing characterisation techniques
3. Review of various ageing classification techniques
4. Accelerated thermal ageing technique
5. Fiber-Optic Sensor
6. Superhydrophobicity and online ageing detection
7. Machine learning models for online ageing detection
8. IoT and online ageing
9. Review of related systems

**Table 2 sensors-22-07923-t002:** Paper Screening Methodology.

Search Index	Specific Content
Research Question	How can ageing detection in transformer oil systems be improved?
Database	RefWorks Proquest, Elsevier Science Direct, IEEE Xplore, Google Scholar, GCU. Library (host to many databases)
Article Type	Scientific articles published in peer-reviewed journals and conferences, technical papers, patents, and generic materials relevant to the field.
Search Strings	HV, Insulator, Ageing, Sensor, Transformer Oil, Superhydrophobicity, IoT
Search Language	English
Research Theme Result Ratio	96 out of 182
Screening Procedure	Relevance to research topic/question(s) judged progressively by the contents of the title, abstract, conclusion/discussion, introduction, and methodology.

**Table 3 sensors-22-07923-t003:** Thematic References.

Themes	References
1. Transformer oil types	[4,18,19,20,21,22,23,24,25,26,27]
2. Review of transformer ageing characterisation techniques	[7,16,17,21,28,29,30,31,32,33,34,35,36,37,38,39,40,41,42,43,44,45,46,47,48,49]
3. Review of various ageing classification techniques	[29,46]
4. Accelerated thermal ageing technique	[7,50,51,52,53,54]
5. Sample ageing detection sensors	[32,38,55,56,57,58,59,60,61]
6. An overview of superhydrophobicity	[62,63,64,65,66,67,68,69,70,71,72,73,74,75,76,77,78]
7. Machine Learning and Uncertainty Quantification	[79,80,81,82,83,84,85]
8. IoT and online ageing	[86,87,88,89,90,91,92,93,94,95,96]
9. Review of related systems	[9,10,57]

**Table 4 sensors-22-07923-t004:** Modified Oil Quality Index [29,46].

S/N	Class	Variable	Range
1.	Good Oil/Class A	NN	0.00 to 0.03
		IFT	45 to 30
		OQIN	1500 to 1000
2.	Proposition A Oil/Class B	NN	0.05 to 0.10
		IFT	29.90 to 27.10
		OQIN	600 to 271
3.	Marginal Oil/Class C	NN	0.11 to 0.15
		IFT	27 to 24
		OQIN	245 to 160
4.	Bad Oil/Class D	NN	0.16 to 0.40
		IFT	23.9 to 18
		OQIN	150 to 45
5.	Very Bad Oil/Class E	NN	0.41 to 0.65
		IFT	17.90 to 14
		OQIN	44 to 22
6.	Extremely Bad Oil/Class F	NN	0.66 to 1.50
		IFT	13.90 to 9
		OQIN	21 to 6
7.	Oil in Disastrous Condition/Class G	NN	1.51 or more
		IFT	8.50 or less
		OQIN	6 or less

**Table 5 sensors-22-07923-t005:** Cross Capacitive Sensor and Fibre Optic Sensor Comparison.

S/N	Sensor Type	Input/Output	Intrinsic/Destructive	Benefit/Limitation
1.	Cross-Capacitive Sensors	Capacitance/Voltage	Yes/No	1. Adaptable for various ageing feature detections. 2. Good Repeatability. 3. Requires a data card (AD7150). 4. It is temperature-independent. 5. Only sensitive to the parameter measured (transformer oil). 6. Prone to electromagnetic interference.
2.	Fibre Optic Sensors	Light/Voltage or Current	Yes/No	1. Potentially easy to install. 2. Allows for offline/online sensing. 3. Resistant to ionising radiation, electromagnetic interference and radio-frequency interference. 4. Explosion-proof. 5. Extended ageing detection applications. 6. Lightweight and high sensitivity.

## Data Availability

Not Applicable.

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
