# Peer review of "Towards Online Ageing Detection in Transformer Oil: A Review"

_sensors, 2022, doi:10.3390/s22207923_

Round 1

Reviewer 1 Report

The manuscript is well presented for the review of online detection of ageing of transformer oil however, the section 9 IOT inception in online ageing detection looks slightly weak and need to be strengthen up.

Author Response

Reviewer 1

Comments and Suggestions for Authors 

The manuscript is well presented for the review of online detection of ageing of transformer oil however, the section 9 IOT inception in online ageing detection looks slightly weak and need to be strengthen up.

Response:

Thanks for your comments/suggestions. Section 10 (previously section 9) has been strengthened by the addition of lines 713 to 722, which lists the elements and tools to enable IoT integration; and the steps and tools for prototyped online data transmission, data pre-processing,  processing, model development, and design of frontends and interfaces (lines 728 to 732) following the framework depicted in Figure 11 of the original manuscript.

Reviewer 2 Report

The paper provides a review of the online ageing detection in transformer Oil . There are just related concerns which authors need to address. 

1.       The paper reviews the online ageing detection methods of transformer oil, including the types of transformer oil, some traditional ageing detection methods and used sensors, while artificial intelligence methods are commonly used to implement ageing detection methods of transformer oil, the related methods are needed to be reviewed in this paper.  

 2.       The manuscript provides the related principle of different sensors, however, the scenarios, advantage and disadvantage of the related sensors are not clearly provided.   

3.       Table of contents is suggested to be added in front of main body of manuscript for easy reading

Author Response

Reviewer 2

Comments and Suggestions for Authors 

1. The paper reviews the online ageing detection methods of transformer oil, including the types of transformer oil, some traditional ageing detection methods and used sensors, while artificial intelligence methods are commonly used to implement ageing detection methods of transformer oil, the related methods are needed to be reviewed in this paper.  

Response:

Thanks very much for pointing this out. A new section (section 9) has been added (lines 619 to 669) which points to some machine learning methods for online ageing detection. In addition, some common off-the-shelf software packages that can be used were also included. Finally, the choice of variables and external references for additional review of each method have been included.

Comments and Suggestions for Authors 

2.  The manuscript provides the related principle of different sensors, however, the scenarios, advantage and disadvantage of the related sensors are not clearly provided.  

Response:

Thanks for your comments and suggestions. A new section on the cross-capacitive sensor has been added (lines 416 to 446), and a table showing the benefits/limitation of the cross-capacitive sensor and fibre optic sensor has been added (lines 571 to 572). In addition, the focus has been placed on intensity-modulated fibre optic sensors, and the comparative advantages/disadvantages of this optical configuration are captured in lines  499 to 517. Finally, a suggestion on overcoming this configuration's limitation is captured in lines 495 to 498.

Comments and Suggestions for Authors 

3. Table of contents is suggested to be added in front of main body of manuscript for easy reading

Response

Thanks for your comments and suggestions. A hyperlinked table of contents has been added to the revised manuscript on lines 27 to 38.

Reviewer 3 Report

The subject is very interesting, and the research is properly conducted. However, I have several suggestions that need to be addressed. They are as follows:

-Keywords should be placed in alphabetical order.

-In the last sentence of the first paragraph of the Introduction section, please add "environmental (ecological) consequences."

-Add more details in figure 1, for example, electrical stress, partial discharge formation, and impurities in the insulation material (transformer oil).

- Add the explanation of partial discharge formation in transformer oil and its effect on the aging process. Additionally, this research should be cited and discussed:

 “Impact of Power Transformer Oil-Temperature on the Measurement Uncertainty of All-Acoustic Non-Iterative Partial Discharge Location”, Materials 2021

-  Also, authors may consider and analyze the following references regarding the aging process of transformer oil:

"Cellulose Degradation and Transformer Fault Detection by the Application of Integrated Analyses of Gases and Low Molecular Weight Alcohols Dissolved in Mineral Oil", Energies 2022

“Aging of Transformer Insulation - Experimental Transformers and Laboratory Models with different Moisture Contents: Part I – DP and Furans Aging Profiles”, IEEE Transaction on Dielectrics 2021

- Line 578, add "and reliability of the implemented AI models.". Additionally, this research should be cited and discussed:

“Application of Machine Learning to Express Measurement Uncertainty”, Applied Science 2022

Author Response

Reviewer 3

Comments and Suggestions for Authors 

1. The subject is very interesting, and the research is properly conducted. However, I have several suggestions that need to be addressed. They are as follows:

 -Keywords should be placed in alphabetical order.

Response:

Thanks for pointing this out. Keywords have been placed in alphabetical order (lines 24 and 25).

Comments and Suggestions for Authors

2.  In the last sentence of the first paragraph of the Introduction section, please add "environmental (ecological) consequences."

Response:

Thanks for pointing this out. The phrase "environmental (ecological) consequences have been added (line 47).

Comments and Suggestions for Authors

3. Add more details in Figure 1, for example, electrical stress, partial discharge formation, and impurities in the insulation material (transformer oil)

Response

Thanks for pointing this out. Figure 1 has been updated to include the effect of electrical stress, partial discharge, and the effect of internal and external solid impurities (line 71)

Comments and Suggestions for Authors

4.   Add the explanation of partial discharge formation in transformer oil and its effect on the ageing process. Additionally, this research should be cited and discussed:

 "Impact of Power Transformer Oil-Temperature on the Measurement Uncertainty of All-Acoustic Non-Iterative Partial Discharge Location", Materials 2021

Response

Thanks for pointing this out. An explanation of partial discharge formation in transformer oil and its effect on the ageing process has been added in lines (59 to 65) and diagrammatically in line (71).

Comments and Suggestions for Authors

5. Also, authors may consider and analyze the following references regarding the aging process of transformer oil:

"Cellulose Degradation and Transformer Fault Detection by the Application of Integrated Analyses of Gases and Low Molecular Weight Alcohols Dissolved in Mineral Oil", Energies 2022

Response

Thank you very much for this reference. It has been cited in section 1 and adequately referenced.

Comments and Suggestions for Authors

6. "Aging of Transformer Insulation - Experimental Transformers and Laboratory Models with different Moisture Contents: Part I – DP and Furans Aging Profiles", IEEE Transaction on Dielectrics 2021

Response

Thank you very much for this reference. It has been adequately added to section 6

Comments and Suggestions for Authors

7.    - Line 578, add "and reliability of the implemented AI models.". Additionally, this research should be cited and discussed:

"Application of Machine Learning to Express Measurement Uncertainty", Applied Science 2022

Response

Thanks for this input. "and reliability of the implemented AI models." has been added to line 687. "Application of Machine Learning to Express Measurement Uncertainty", Applied Science 2022 reference has been added and cited. A discussion on the need for uncertainty quantification and a detailed review reference has been added to lines 662 to 669.

Round 2

Reviewer 2 Report

I feel the paper is much improved and the responses have been integrated into the paper well.

Reviewer 3 Report

The authors have answered all the comments of my review. The paper can be published in its current form.